# Advances in Evaluation of Antioxidant and Toxicological Properties of *Stryphnodendron rotundifolium* Mart. in *Drosophila melanogaster* Model

**DOI:** 10.3390/foods11152236

**Published:** 2022-07-27

**Authors:** Gerson Javier Torres Salazar, Assis Ecker, Stephen Adeniyi Adefegha, José Galberto Martins da Costa

**Affiliations:** 1Postgraduate Program in Ethnobiology and Nature Conservation, Regional University of Cariri, Coronel Antônio Luíz Street, 1161-Pimenta, Crato 63105-010, CE, Brazil; timotygertor@yahoo.com; 2Postgraduate Program in Toxicological Biochemistry, Department of Biochemistry and Molecular Biology, Center of Natural and Exacts Sciences, Federal University of Santa Maria, Santa Maria 97105-900, RS, Brazil; assisecker@hotmail.com (A.E.); saadefegha@futa.edu.ng (S.A.A.); 3Functional Foods, Nutraceuticals and Phytomedicine Laboratory, Department of Biochemistry, Federal University of Technology, Akure 340001, Nigeria; 4Postgraduate Program in Biological Chemistry, Regional University of Cariri, Coronel Antônio Luíz Street, 1161-Pimenta, Crato 63105-010, CE, Brazil

**Keywords:** *Stryphnodendron rotundifolium* Mart., antioxidant activity, iron overload protection

## Abstract

This study investigated the flavonoid content, antioxidant activity, and toxicological properties of the acetone–water fraction of stem bark of *Stryphnodendron rotundifolium* Mart. (TFSR). The total flavonoid content and antioxidant activity were determined, as typified by DPPH^●^ and ABTS^●+^ radical scavenging abilities, Fe^3+^ reducing antioxidant power (FRAP), relative antioxidant capacity (RAC), and the inhibition of thiobarbituric acid reactive species (TBARs) in *Drosophila melanogaster* tissue. Toxicity and locomotor functions were evaluated in adult *D. melanogaster* flies through aging and survival assays, startle-induced negative geotaxis, and centrophobic responses with video-assisted open field motion tracking. The flavonoid content of dry TFSR (DF) was 3.36 mg quercetin/g. Furthermore, the significant antioxidant activity of TFSR was revealed through scavenging 95.3% of the ABTS^●+^ radical and 82.4% of the DPPH^●^ radical, as well reducing 74.7% of Fe^3+^ in the FRAP assay and 80% Mo^6+^ in the RAC assay. TFSR conferred 70.25% protection against lipid peroxidation in *Drosophila* tissue. Survival rates ranged from 84.65 to 103.98% in comparison to the non-supplemented control and no evident deterioration of locomotor functions and centrophobia responses was observed. These results revealed that TFSR has potent antioxidant activity and low toxicity in vivo, profiling TFSR as a promising natural product in the treatment/management of iron overload and associated conditions.

## 1. Introduction

*Stryphnodendron rotundifolium* Mart, popularly known as “Barbatimão”, is widely used in Brazilian traditional medicine for the treatment of wounds, ulcers, gastritis, inflammation, and vaginal infections. The stem bark parts are mainly used for the preparation of teas, topic solutions and sitz baths, and decoction. They can also be prepared by infusion in water. The reports of dosages in the preparation of medicinal plants are scarce, and there is uncertainty in the dosage that can be administered in vivo [1,2,3,4,5,6].

Many studies have demonstrated that a variety of chronic diseases result from an imbalance between the generation of reactive oxygen species (ROS) and antioxidant mechanisms in host cells, providing evidence for the therapeutic potential of antioxidant compounds [7]. Previous studies have attributed the antioxidant activity of *S*. *rotundifolium* Mart, to the presence of a matrix of phytochemicals capable of inhibiting the generation of (ROS) and chelating Fe^2+^ overload [8,9]. It is consistently evident that phenolic compounds, including flavonoids and tannins, are crucially responsible for the antioxidant activity of natural products derived from this species [8,9].

Regarding the toxicity of extracts from these species, previous reports have shown controversial results. Oral administration of the water:ethanol extract (1:1; *v*/*v*) of *S. rotundifolium* stem bark at a dose of 400 mg/kg in *Wistar* rats caused progressive damage to liver function [10]. On the contrary, it was shown that dry ground husks and the ethanol:water extract (1:1; *v*/*v*) of husks mixed with a diet based on soybean bran, corn, oat hay, and protein supplement did not cause liver damage in confined lambs [11]. The acetone:water extract (7:3; *v*/*v*) of the dried stem bark of *S. rotundifolium* 250 μg/mL showed low toxicity (<10%) in the *Artemia Salina* assay [9]. Gallic acid, monomeric PAs, dimeric and trimeric compounds with gallate substitutions, and C-hexosyl-O-pentosyl-5,7-dihydroxychromone isomer were identified in the extract [9]. In addition, the water:ethanol extract (1:1; *v*/*v*) of bark exhibited a cytoprotective effect on plants against Al^3+^ and Hg^2+^ [12].

These divergent views may be due to errors in the standardization of appropriate concentrations from medicinal preparations and detailed characterization of the main secondary metabolites that contribute to biological activities with minimal toxicity. Therefore, the evaluation of both the toxicological effects and antioxidant activities of the stem bark of the species is highly desirable to ensure its safe usage. The present study was a breakthrough in the investigation of in vitro antioxidant activity and in vivo toxicity at therapeutic concentrations of the tannin-rich fraction of *S. rotundifolium* (TFSR), using *Drosophila* melanogaster as an experimental model.

## 2. Materials and Methods

### 2.1. Botanical Material and Extraction Procedure

*Stryphnodendron rotundifolium* Mart. stem bark samples were collected in the Araripe National Forest in the municipality of Crato, Ceará, Brazil (coordinates: 07°81′ S, 039°28′ W). Dr. Maria Arlene Pessoa identified the species, and a voucher specimen was prepared and registered at the Herbarium of the Regional University of Cariri (registry number: 14,074).

The tannin-rich fraction of the stem bark of *Stryphnodendron rotundifolium* Mart. was obtained according to the method described by [9]. Freshly collected barks were dried, crushed, and subjected to fat removal with hexane for 24 h. Then, hexane was removed from the solid material at 70 °C. Following this process, the material was subjected to exhaustive maceration using a 7:3 acetone–water mixture for 72 h at room temperature to avoid interactions between phenolics and vegetable proteins, favoring the obtainment of more stable phenolic compounds. This procedure was repeated three times as previously described. The resulting solution was subjected to the exhaustive removal of acetone at 65 °C and reduced pressure, until the volume was reduced to one-third of the initial extractive mixture, after which it was concentrated in a water bath at 70 °C for 24 h. The acetone-free concentrated liquid was frozen and subjected to lyophilization, after which the tannic fraction was obtained as an amorphous solid.

### 2.2. Total Flavonoid Content

The total flavonoid content was determined according to the method proposed by [13]. A stock solution was prepared using 0.025 g of extract in 10 mL of methanol (70% *v*/*v*). An aliquot of 2 mL of this solution was added to 1 mL of AlCl_3_ (15% *v*/*v*) and the volume was adjusted to 10 mL using methanol (70% *v*/*v*). After 30 min, the absorbance was read at 425 nm. The flavonoid content was calculated by linear regression using the quercetin calibration curve (0.001–0.012 mg/mL). The experiments were performed in triplicate and the results were expressed in mg of quercetin (Q)/g dry fraction (DF).

### 2.3. Stock of Flies

*Wild-type D. melanogaster* (Canton-S strain) was obtained from the National Species Stock Center, OH, USA, grown in flasks measuring 6.5 × 6.5 × 7.0 cm (approximate volume of 300 mL) and kept in a humidified incubator (60%) with controlled temperature at 24 °C and a 12 h on/off light cycle. The standard fly colony was maintained on a diet containing a mixture of 75% corn flour, 10% sugar, 6% soy flour, 2.5% wheat germ, 5% powdered milk, and 1.5% salt. A pinch of lyophilized yeast and 0.08% of methyl paraben (p-hydroxybenzoate) were added to prevent fungal growth.

#### Whole Fly Homogenate Preparation

Briefly, ice-anesthetized flies were separated and manually homogenized in 50 mM Tris-HCl buffer pH 7.0, at 0.04605 g (approx. 76 flies)/500 μL; the homogenate was centrifuged at 564 × *g* for 10 min at 4 °C. After centrifugation, the supernatant (S_1_) was kept at 4 °C on ice until assays.

### 2.4. Analysis of In Vitro Antioxidant Activity

#### 2.4.1. ABTS^•+^ Free Radical Capture

The ABTS^•+^ scavenging activity of the fraction was analyzed using the methodology described by [14] with adaptations. An aqueous solution (7 mM) of 2,2′-azino-bis(3-ethylbenzothiazoline-6-sulfonic acid) (ABTS) was mixed with 2.45 mM potassium persulfate and kept at room temperature in the dark for 16 h to generate ABTS^•+^ free radicals. Afterwards, the solution was diluted with 150 mM phosphate buffer (pH 7.3) to adjust the absorbance between (0.700 ± 0.020–0.800 ± 0.020) and read at 734 nm at 30 °C. Samples (30 μL) of TFSR or quercetin (Q)/rutin (Rut) standard solutions at concentrations in the range (0.5–5.0) μg/mL were added to 2970 μL of ABTS^•+^ solution and 5 min later the absorbance was measured at 734 nm. The extent of discoloration was expressed as percentage (%) of inhibition of the ABTS^•+^ radical according to Equation (1):(1)$$\mathrm{Inhibition}\left(\%\right)=\frac{[\left({\mathrm{Abs}}_{\mathrm{Control}}-\left({\mathrm{Abs}}_{\mathrm{Sample}}-{\mathrm{Abs}}_{\mathrm{Blank}}\right)\right]\times 100\%}{{\mathrm{Abs}}_{\mathrm{Control}}}$$
where Abs_Control_ is the absorbance of the ABTS^•+^ radical solution, Abs_Sample_ is the absorbance of the ABTS^•+^ radical solution after 5 min of the addition of the fraction or the positive control patterns (quercetin or rutin) at the different concentrations, and Abs_Blank_ is the absorbance of the sample, quercetin, or rutin without ABTS^•+^ radical solution.

#### 2.4.2. Iron-Reducing Antioxidant Power (FRAP)

The FRAP method was used to evaluate the ability of the extract to reduce Fe^3+^, as previously described by [15,16]. Briefly, 300 μL of TFSR (0.25 to 5.0 μg/mL) was mixed with 3975 μL of FRAP reagent containing 0.3 M acetate buffer pH 3.6, (10 mM 2,4,6-Tripyridyl-s-triazine TPTZ in HCl 40 mM) and 20 mM FeCl3 at 10:1:1 (*v*/*v*/*v* ratio) with 225 mL of methanol. After a 4 min incubation at room temperature and protected from light, the absorbance was measured at 595 nm. Reduction curves in the range (0.25–3.0) μg/mL of gallic acid (GA), quercetin (Q), and catechin (Cat) standards were also developed.

#### 2.4.3. DPPH^●^ Assay

The 1,1-diphenyl-2-picrylhydrazyl (DPPH^●^) scavenging activity was determined according to the protocol described by [17] with slight modifications. A total of 20 μL of the fraction or ascorbic acid (AA, control) in concentrations ranging from 0.001 to 0.25 mg/mL were resuspended in 80 μL of 95% ethanol and 100 μL of 0.3 mM DPPH^●^ solution in ethanol. Readings were taken at 518 nm after a 30 min incubation at room temperature and in the absence of light. The DPPH^●^ free radical inhibiting capacity of the fraction was expressed as a percentage of the inhibition caused by the control (100 μL of 0.3 mM ethanolic DPPH^●^ and 100 μL of 95% ethanol) and calculated using the following Equation (2):(2)$$\mathrm{DPPH}\;\mathrm{inhibition}\;\left(\%\right)=\frac{[{\mathrm{Abs}}_{\mathrm{Control}}-\left({\mathrm{Abs}}_{\mathrm{Fraction}}-{\;\mathrm{Abs}}_{\mathrm{Blank}}\right)]\times 100\%\;}{{\mathrm{Abs}}_{\mathrm{Control}}}\;$$
where Abs_Control_ corresponds to the absorbance of the DPPH^●^ solution + ethanol, Abs_Fraction_ is the absorbance of the fraction + DPPH^●^, and Abs_Blank_ corresponds to the absorbance of the fraction + ethanol.

#### 2.4.4. Relative Antioxidant Capacity (RAC)

To assess the relative antioxidant capacity, 100 μL of varying concentrations of TFSR (0.001–0.025 mg/mL) were added to 1000 μL of 10.8 mM ammonium molybdate (reagent solution), 1000 μL of 75.6 mM sodium phosphate buffer (pH 7.4), 700 μL of 1.62 M sulfuric acid, and 200 μL of milli-Q water. Each sample was incubated in a water bath at 95 °C for 90 min. After cooling to room temperature, readings were taken at 695 nm. Rutin (0.075 mg/mL), catechin (0.020 mg/mL), and ascorbic acid (0.020 mg/mL) were used as standards. The relative antioxidant activity (3) was calculated by Equation (3), as shown below [18].
(3)$$\mathrm{RAC}\;\left(\%\right)=\frac{\left[\left({\mathrm{Abs}}_{\mathrm{Fraction}}-{\;\mathrm{Abs}}_{\mathrm{Fraction}\;\mathrm{blank}}\right)\times 100\%\right]}{\left({\mathrm{Abs}}_{\mathrm{Control}}-{\;\mathrm{Abs}}_{\mathrm{Control}\;\mathrm{blank}}\right)}$$

Abs_Fraction_ = absorbance of the fraction + reagent solution, Abs_Fraction blank_ = absorbance of the fraction, Abs_Control_ = absorbance of the standard + reagent solution, Abs_Control blank_ = absorbance of the standard.

#### 2.4.5. Inhibition of Lipid Peroxidation

The inhibition of lipid peroxidation was evaluated by measuring the levels of thiobarbituric acid reactive substances (TBARs) of the fraction, as described by [19] with modifications. Briefly, 150 μL of different concentrations of TFSR and positive controls catechin, quercetin, and ascorbic acid were pre-incubated at 37 °C for 1 h with a mix containing (S_1_) (320 μL), 50 mM phosphate buffer pH 7.4 (450 μL), 500 μM EDTA disodium salt (120 μL), 8.1% *w*/*v* sodium dodecyl sulfate SDS (200 μL), 60 μL of deionized water and 200 μL Fe^2+^ (final concentration of 55 μg/mL) to induce peroxidation. Afterwards, 0.8% TBA (750 μL) and 2.8% TCA (750 μL) were added and incubated for 2 h at 100 °C. After this phase, the samples were cooled to room temperature, centrifuged at 2422 RCF, and the absorbance was measured at 532 nm. The inhibition of peroxidation was calculated by Equation (4).
(4)$$\mathrm{Inhibition}\;\mathrm{of}\;\mathrm{peroxidation}\;\left(\%\right)=\frac{[\left({\mathrm{Abs}}_{\mathrm{Control}}-\left({\mathrm{Abs}}_{\mathrm{Fraction}}-{\;\mathrm{Abs}}_{\mathrm{Blank}}\right)\right]\times 100\%}{{\mathrm{Abs}}_{\mathrm{Control}}}\;$$

Abs_Control_ = absorbance (S_1_ + Fe^2+^), Abs_Fraction_ = absorbance (S_1_ + Fe^2+^ + fraction) and Abs_Blank_ = absorbance (Fe^2+^ + fraction).

### 2.5. In Vivo Toxicity Assays

#### 2.5.1. Aging and Survival Assays

*Drosophila melanogaster* flies (1–2 days old) of both sex were randomly separated and treated on standard food of the lyophilized yeast-free colony supplemented with TFSR at [2.000, 1.000, 0.500, 0.250, 0.200, 0.100, 0.050, and 0.000 (control)] mg/g diet.

Survival rate was assessed according to [20], transferring flies to fresh food media every 24 h and recording the mortality. Individual flies were censored from the results if they escaped during tipping on transfer or if they died from a cause unrelated to age or getting stuck in food. This process was followed until reaching the time when each concentration killed 50% of each population of flies (LD_50_) and followed until the majority of flies died. Approximately 300 flies per group were included in the survival experiments. The total number of flies represents the sum of three independent experiments (approximately 100 flies for each treatment repetition). The LD_50_ of each concentration of TFSR was determined.

#### 2.5.2. Startle-Induced Negative Geotaxis

After reaching the LD_50_ of each supplemented concentration of TFSR, the locomotor performance of the flies was evaluated by the negative geotaxis test at 25 °C according to [21,22,23]. Thus, 3 × (10–15) surviving flies were separated under brief ice anesthesia (4 min) and then transferred to glass columns in an upright position (length: 30 cm, diameter: 1.5 cm) and capped with cotton. After 15 min of recovery from the cold exposure, the flies were gently tapped at the bottom of the column and the number of flies able to climb to the top of 30 cm was recorded n_top_. Flies that reached the top of the column and those that remained on the bottom n_bottom_ were counted separately after 60 s. Assays were repeated three times at 5 min intervals. The scores are the average of the number of flies on top (n_top_) and on the bottom (n_bottom_), expressed as percentages of the total number of flies (n_total_). For each experiment, a performance index (PI) was calculated, defined as 1/2 [(n_total_ + n_top_ − n_bottom_)/n_tot_] [24]. If all flies climb to the top of the tube, the score is 1, and if no flies climb, the score is 0. Results are shown as the mean ± SEM of scores obtained in three independent experiments (*n* = 3) in triplicate.

#### 2.5.3. Video-Assisted Open Field Motion Tracking

After reaching the LD_50_ of each supplemented concentration of TFSR, the open field motion tracking assay was performed as described by [22] with some modifications. The fly tracking space was confined by lids of Petri dishes (15 cm diameter × 1.5 cm tall), so that the flies could walk and breathe freely and normally. The plates were supported on surfaces with graph paper at a controlled room temperature of 25 °C. A portable HD digital camera with a resolution of 1080 pixels was positioned 15 cm above the Petri dishes and connected to an HP Pavilion TouchSmart 11 Notebook PC for video processing. A concentric frame of 9 × 9 cm (81 cm^2^) was delimited on the graph paper. After reaching the LD_50_ of each TFSR concentration, groups of 15–20 flies in triplicates were chilled on ice for 3–4 min before being placed in the center of the cubic concentric frame in the space confined by the plates of Petri and moved under the camera to record 5 min long tracking videos.

The movement exploration of each group of flies was calculated by dividing the plate into 150 × 150 lines, equivalent to approximately 17,671 boxes (π × (75 mm)^2^). It was calculated the (%) of flies that were 75% away from the center of the plate within 10 min. Mean walking activity scores for each treatment group were determined for three independent experiments (*n* = 3) and expressed as (%) centrophobia ± mean standard deviation.

Centrophobia was represented as the amount of time that each fly remained in a region 75% from the center of the plate. Smaller time values indicate that the fly is more centrophobic. In the case of the 15 cm plate, this is an area of 132.53 cm^2^ (r = 6.495) in a total area of 176.7 cm^2^. The XY coordinates were zeroed to the center of the plate and converted to polar coordinates to use the radial coordinate as a measure of distance from the center. The percentage of time in the central region was calculated as the time that the flies spent in the area below 75% of the center divided by the total time (10 min).

### 2.6. Statistical Analysis

The results obtained on the antioxidant properties and toxicity effect assays were expressed as the means ± standard error of mean (SEM) of at least three experiments and analyzed with a one-way ANOVA followed by Tukey’s post hoc test for multiple comparisons of data with normal distribution and similar standard deviation using the GraphPad Prism software version 6.0. Statistical significance was considered when *p* < 0.05.

## 3. Results and Discussion

### 3.1. Total Flavonoid Content and In Vitro Antioxidant Activity ABTS^•+^, FRAP, DPPH^●^, RAC

The concentration of flavonoids was 3.36 ± 0.09 mg quercetin/g dry TFSR (DF). Taking into account the report from the previous work conducted by our research group [9], on the same fraction (TFSR) we were able to quantify condensed tannins (54.83 ± 2.30 mg catechin equivalent/g DF) and hydrolysable tannins (28.84 ± 2.21 mg tannic acid equivalent/g DF) with a predominance of flavan-3-ol and type B proanthocyanidins (PAs) with C_4_→C_8_ and C_4_→C_6_ interflavanic bonds with a high degree of hydroxylation, some of these compounds including 4′-O-methyl-epigallocatechin-(4→8)-epigallocatechin, epigallocatechin-(4β→8)-epigallocatechin, epigallocatechin-(4→8)-epigallocatechin-(4→8)-epigallocatechin, 4′-O-methyl-epigallocatechin-3-O-gallate-(4→6)-epigallocatechin-3-O-gallate, epigallocatechin-(4→8)-epigallocatechin-(4→8)-epigallocatechin-3-O-gallate, epigallocatechin-(4→8)-epigallocatechin-3-O-gallate, and robinetinidol-4′-O-methyl-(4→8)-epigallocatechin. Flavonoids were identified as the compounds with the least concentration in TFSR.

Therefore, the presence of flavonoids may have exerted a minor influence on the considerable antioxidant activity of TFSR through reducing and anti-radical action. Results on the in vitro antioxidant capacity assessed by the ABTS^•+^ radical scavenging ability assay, the reducing action of Fe^3+^ in the FRAP assay, and the DPPH^●^ radical scavenging ability are presented in Table 1. TFSR scavenged ABTS^•+^ radical in a dose-dependent manner (R^2^ 0.98), exhibiting better radical scavenging ability than the standard compound rutin (Table 1). TFSR reached a maximum scavenging ability of 95.3 ± 0.6% at a concentration of 5.0 μg/mL. However, it was less potent than the standard quercetin, recognized as the flavonoid with the highest ROS scavenging activity in vitro [25].

Based on the extract concentration that caused 50% inhibition (IC_50_), TFSR showed potent DPPH^•^ radical scavenging activity (IC_50_ = 8.65 ± 1.10 μg/mL) with a maximum inhibition (82.4 ± 3.2%), observed at a concentration of 250 μg/mL, similar to the ascorbic acid standard (Table 1). As shown in a previous study [8], the hydroalcoholic and aqueous extracts of the bark of *S. rotundifolium* Mart, had the IC_50_ values of 5.4 and 12.0 μg/mL respectively. In this study, a similar trend was observed for TFSR as well. However, the extracts showed weak Fe^2+^ chelating activity. Furthermore, the fraction showed a concentration-dependent Fe^3+^ reducing ability in the [Fe(III) (TPTZ)_2_]^3+^ complex (R^2^ 0.98), reaching a maximum value of 74.7 ± 0.2% at a concentration of 5.0 μg/mL, in comparison to the standard substances gallic acid, quercetin, and catechin (Table 1). These results may be due to the lower steric impediment between the standard molecules and the three-dimensional arrangement of the [Fe(III) (TPTZ)_2_]^3+^ complex in relation to the set of larger molecules of flavan-3-ol and polymeric proanthocyanidins (PAs) predominant in TFSR. This may have led to a lower ability to influence Fe^3+^ and consequently exert a greater reducing effect.

The fraction exhibited RAC values ranging from 71.62 to 80.62% in relation to the 100% activity of the ascorbic acid, rutin, and catechin standards and IC_50_ values between 0.0111 to 0.01296 mg/mL (Table 2). These RAC results, together with the Fe^3+^ reducing ability of TFSR in the FRAP assay, indicate that *S. rotundifolium* is a species with potent antioxidant agents capable of reducing metal ions. Evidence has shown that flavan-3-ol with catechol or pyrogallol groups at the B ring, in the presence of π-conjugated structures allow for electronic shift and provide a greater capacity to reduce metal ions and scavenge DPPH^●^ radicals that can be stabilized by resonance to generate semiquinones and quinones through electron donation and proton release [26,27,28,29]. Thus, the extracts inhibited DPPH^●^ and ABTS^•+^ radicals as well as reduced Mo^6+^ and Fe^3+^ ions due to the release of protons and electrons by polymeric PAs and flavan-3-ol present in TFSR, as reported in a previous study carried out by our research group [9]. These biological effects could also be attributed to the presence of prodelphinidins, procyanidin, and prorobinetinidine, which have been previously identified [9].

The remarkable ABTS^•+^ and DPPH^•^ sequestering activity of the fraction is associated with the presence of hydrolysable tannins and mainly B-type proanthocyanidins (PAs) (dimers and trimers) which contain catechol and pyrogallol groups in their structures. These compounds have been recognized as excellent natural antioxidants and, as such, can contribute to the prevention of various diseases including cancer, diabetes, obesity, and neurodegenerative diseases [30]. Recent studies indicate that the mechanisms underlying the antioxidant activity of these types of compounds of the genus *Stryphnodendron* involve the donation of hydrogen and electrons responsible for the inhibition of free radicals and reduction of metal ions [9,31]. Furthermore, evidence indicates that the higher the degree of hydroxylation, the stronger the antioxidant power of these compounds [32]. The optimization of antioxidant activity is also justified by the presence of o-hydroxyl groups, which may increase the degree of galloylation. The DPPH^•^ scavenging activity of proanthocyanin polymers has been shown to increase from monomers to tetramers, although in procyanidins with a high molecular weight, tetramer activity may be impaired due to steric hindrance. A similar phenomenon can occur when galloyl groups bind to each other, preventing the reaction with free radicals or metal ions such as Mo^6+^, Fe^3+^, and Fe^2+^ [33].

#### Lipid Peroxidation Inhibition

Regarding the ability to protect against in vitro membrane lipid peroxidation of *Drosophila melanogaster* homogenate, TFSR notably protected against peroxidation with IC_50_ of 29.76 ± 5.71 μg/mL and maximum inhibition activity of 70.25 ± 4.59% at 0.075 mg/mL. At a concentration of 0.050 mg/mL, TFSR exhibited inhibition of 60.58 ± 4.74%, statistically similar to the catechin control (53.62 ± 6.17%), whereas at a concentration of 0.025 mg/mL, it inhibited peroxidation at 46.35 ± 4.76%, statistically similar to the quercetin control (45.67 ± 4.19%) but significantly higher than the ascorbic acid control (35.40 ± 4.30%) (*p* < 0.05) (Figure 1).

The inhibition of lipid peroxidation induced by Fe^+2^ overload (55 μg/mL) in *Drosophila melanogaster* homogenate at all concentrations of TFSR tested and Fe^2+^ chelating ability of TFSR may contribute towards the reduction in Fe overload. However, the observed ability of TFSR to scavenge DPPH^●^ and ABTS^●+^ radicals also supports the hypothesis that there is a contribution of scavenging lipoperoxyl radical (LOO^●^) commonly found in biological substrates and less reactive than the HO^●^ radical [34]. TFSR inhibits oxidative degradation in *Drosophila melanogaster* tissue exposed to Fe^2+^ overload, mainly due to the chelating action of the proanthocyanidin constituents of TFSR. Therefore, TFSR could be suggested for the development of bioproducts that help to combat oxidative damage mediated by iron overload. The significant free radical scavenging activity, metal ion reduction, and inhibition of lipid peroxidation by TFSR is related to synergistic effects of the present prodelphinidins, prorobinetidins, and procyanidins, according to previous identifications of our research group [9], indicating that *S. rotundifolium* Mart. possesses the potential to be used in the development of target drugs for the treatment of diseases associated with Fe overload and oxidative stress-mediated diseases. High concentrations of Fe^2+^ play an important pathophysiological role, which include a wide variety of clinical disorders such as secondary hemosiderosis related to the recurrent transfusion of packed red blood cells that lead to iron overload, thalassemia, sickle cell anemia, refractory aplastic anemia, myelodysplastic syndromes, pure erythroid series aplasia, and acute leukemias [35,36,37].

### 3.2. Toxicity and Locomotor Functions

#### 3.2.1. Survival Rate

*D. melanogaster* is considered an organism that grows synchronously, and putative antioxidant molecules affect survival rate and/or locomotor activity [38]. Toxicity through the aging and survival assay of *D. melanogaster* flies supplemented with TFSR revealed that the fraction did not show any toxicity against *D. melanogaster* in the range of 0.050–0.250 mg/mL when compared to the control for the first 28 days of supplementation. It was only from days 29 and 30 that TFSR (0.050, 0.100 and 0.200 mg/mL) exhibited significantly higher toxicity than the control. The toxicity of TFSR was significantly higher when compared to the control on days 9 and 10 at 1.000 and 2.000 mg/mL, respectively. However, the toxicity of the 0.500 mg/mL supplementation was significantly higher than the control on day 14 (Figure 2).

Toxicity (<10%) of 250 μg/mL TFSR in *Artemia salina* has already been reported [9]. Due to the low toxicity of TFSR in the range of 0.050–0.250 mg/mL, it can be inferred that TFSR may represent a potentially safe therapeutic agent that can either be used by oral ingestion or in the form of a possible standardized bio-product.

#### 3.2.2. Locomotor Functions

The locomotor function evaluated by the negative geotaxis induced by startle assay in *D. melanogaster* supplemented with TFSR 50.00–2000.00 μg/mL for up to 19–35 days was not significantly affected when compared to the control (Figure 3). *D. melanogaster* shows age-related decline in startle-induced locomotion and negative geotaxis [22,39,40]. As observed in Figure 3, it was observed that the performance index, a measure of the adhesion capacity and speed in the climbing of the flies, was in the range of 0.78–0.92 with no significant difference in relation to the control (0.86). This indicates that the concentration of the TSFR fraction administered to *D. melanogaster* did not significantly alter the locomotor parameter of *D. melanogaster*.

Genetic tools combined with behavioral analyzes have allowed for the investigation of brain structures involved in locomotor activity in vertebrates and invertebrates. In *Drosophila melanogaster*, complex integrated behaviors depend on locomotion for their execution. In this case, video tracking allows for the study of several parameters of locomotor activity, revealing that *Drosophila melanogaster* presents centrophobism and/or thigmotaxis, a behavior related to the spatial orientation of avoiding central zones and staying in the periphery. This behavior is impaired when adenosine monophosphate transduction pathways (cAMP) are disturbed and implicated in olfactory learning and memory within mushroom bodies (MBs) [41,42]. The video-assisted open-field motion tracking revealed that centrophobia was not significantly diminished in *Drosophila melanogaster* supplemented with TFSR (2000.00–50.00 μg/mL) as demonstrated in Figure 4, indicating rates of centrophobia in the range of 81–97% for all concentrations without significant differences when compared to the control values (88%). In all concentrations, constant, harmonic and tremor-free flight and locomotion movement were observed.

The antioxidant effects demonstrated by the fraction may be attributed at least partially to the presence of constituents with pyrogallol groups containing hydroxyls that are capable of promoting both Fe^3+^ reduction and Fe^2+^ chelation, in addition to potentially acting as electron and hydrogen donors. According to [43], the presence of galloyl moieties as well as the stabilization of phenoxyl radicals by resonance, contribute significantly to the antioxidant capacity of phenolic compounds. The PAs found in TFSR have been well known to enhance antioxidant action. Furthermore, these compounds, when present in low degrees of polymerization, can be easily absorbed in the gastrointestinal tract, which is a desirable feature for substances used in the development of oral drugs [44]. Despite the strong influence of pyrogallol and phenolic hydroxyl groups on the expressive antioxidant activity of TFSR, it is possible that three-dimensional arrangements of larger PAs generate spatial impediments in the action of radical scavenging and metal ion chelation. Thus, both the dimers and trimers of PAs presents in TFSR may have worked synergistically or additively and contribute towards the antioxidant activity of the fraction.

Iron has a unique ability to alter its redox balance in response to ligands, crucial to many cellular processes. For this reason, cells maintain the concentration of free iron at the minimum level necessary to avoid its toxic effects [45]. Evidence suggests that substances capable of forming complexes with iron and promoting its excretion may have applications in the treatment of various pathological conditions [46,47]. For iron overload in blood fluids, Deferasirox (DFX), an oral iron chelator, was developed. Its main use is to reduce chronic iron overload in patients receiving long-term blood transfusions for conditions such as beta-thalassemia and other chronic anemias. However, renal failure and cytopenias have been reported in patients receiving DFX oral suspension tablets [48]. DFX treatments have shown feasibility and efficacy in patients with thalassemia intermedia [49], myelofibrosis (MF), myelodysplastic syndrome (MDS) [50,51,52], and myeloproliferative neoplasms in fibrotic phase (MPN) [53]. Due to reports of serious adverse reactions, DFX was discontinued in 2020; however, it is available as a generic drug [54]. Studies have shown that extracts with high concentrations of phenolic compounds inhibit lipid peroxidation in human erythrocytes by decreasing the production of malondialdehyde (MDA) [55,56]. Therefore, the fraction rich in polyhydrosylated proanthocyanidins (PAs) obtained from *S. rotundifolium* Mart. (TFSR) may represent a source of potent inhibition of lipid peroxidation, metal ion reducer, antiradical and chelator of Fe^2+^ in overload. Through oral ingestion mixed with meals, it could help to inhibit the initiation and propagation of reactions of free radical formation and the consequent production of malondialdehyde (MDA) in pathological conditions characterized by Fe^+2^ overload. However, further studies are required on TFSR in experimental clinical models to validate the results obtained in vitro.

## 4. Conclusions

The present study demonstrated that TFSR, composed mainly of high hydroxylated proanthocyanidins (PAs) and a low flavonoid content, exerted antioxidative capacity through its ability to scavenge free radicals, reduce and chelate metal ions, and contribute to the inhibition of lipid peroxidation under Fe^+2^ overload conditions at concentrations that do not cause significant toxicity in terms of survival time nor changes in locomotor functions in adult *Drosophila melanogaster*. Furthermore, TFSR exhibited potent antioxidant activity with low toxicity in vivo and, as such, possesses the potential to be used in the development of bioproducts by oral ingestion with therapeutic purposes for the treatment of Fe overload and oxidative stress mediated diseases. Further studies are required using different in vivo and clinical models to validate the results from this study and other studies on the plant.

## Figures and Tables

**Figure 1 foods-11-02236-f001:**
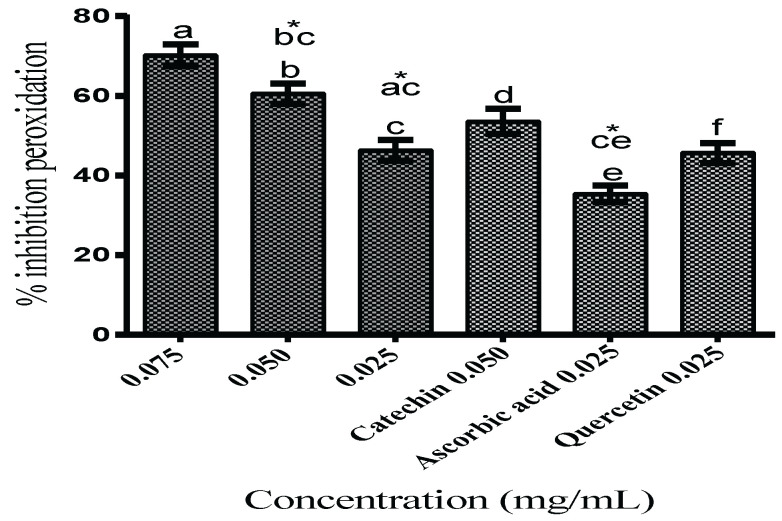
Inhibition of lipid peroxidation in homogenate of whole *Drosophila melanogaster* flies exposed to Fe^2+^ overload and treated with TFSR at different concentrations. The values are expressed as means ± S.E.M (*n* = 3) as calculated by one-way ANOVA followed by Tukey’s multiple comparisons test with a single pooled variance (*p* < 0.05; 95% confidence interval). Bars with letters a, b, and c represent TFSR concentrations, and d, e, and f represent catechin, ascorbic acid, and quercetin concentrations, respectively. “*” indicate significant differences between TFSR concentrations and control substances.

**Figure 2 foods-11-02236-f002:**
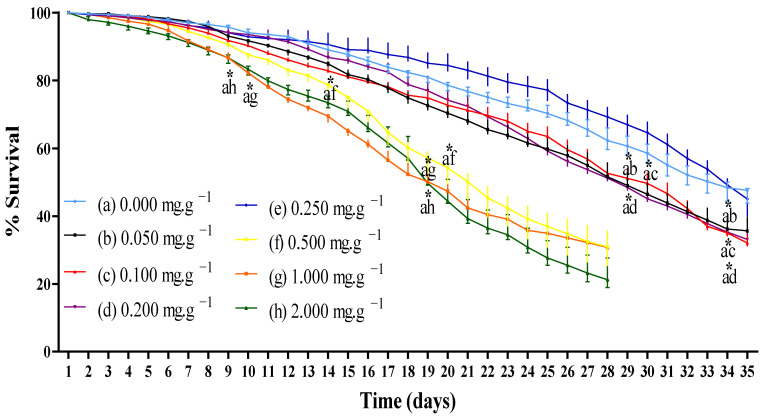
Survival curve of *Drosophila melanogaster* flies exposed to TFSR in different concentrations. The values are expressed as means ± S.E.M (*n* = 3) as calculated by one-way ANOVA followed by Tukey’s multiple comparisons test, with a single pooled variance (*p* < 0.05; 95% confidence interval). Letters a, b, c, d, e, f, g, and h represent concentrations of TFSR and (*) asterisks indicate significant differences between concentrations in the same timepoint (day).

**Figure 3 foods-11-02236-f003:**
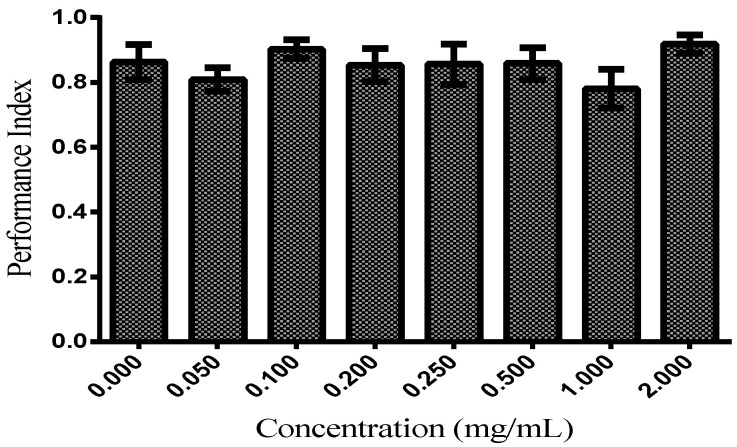
Performance index of the *Drosophila melanogaster* in the geotaxis negative assays. The values are expressed as means ± S.E.M (*n* = 3) as calculated by one-way ANOVA followed by Tukey’s multiple comparisons test with a single pooled variance (*p* < 0.05). No significant differences between concentrations.

**Figure 4 foods-11-02236-f004:**
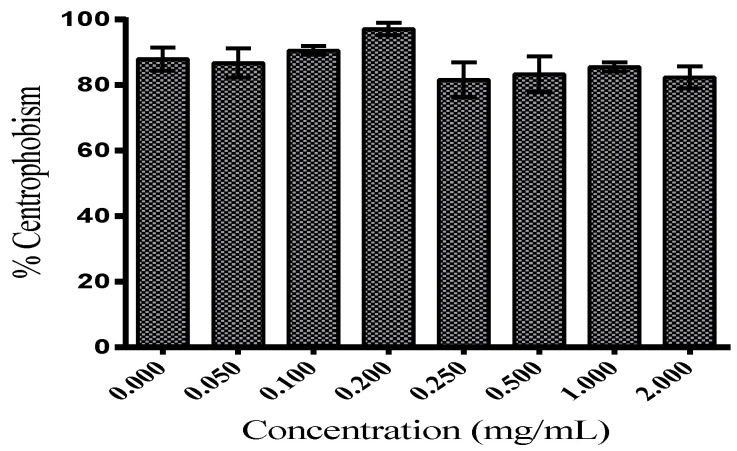
Percentage (%) of centrophobism of the *Drosophila melanogaster* in the open field test. The values are expressed as means ± S.E.M (*n* = 3) as calculated by one-way ANOVA followed by Tukey’s multiple comparisons test with a single pooled variance (*p* < 0.05). No significant differences between concentrations.

**Table 1 foods-11-02236-t001:** Reducing and anti-radical activity of the acetone–water fraction from the stem bark of *Stryphnodendron rotundifolium* Mart (TFSR).

	ABTS^•+^	FRAP	DPPH^●^	ABTS^•+^	FRAP	DPPH^●^
	(IC50 ± SEM) μg/mL			(Max. Act. ± SEM)%		
TFSR	0.74 ± 0.12	3.27 ± 0.13	8.65 ± 1.10	95.30 ± 0.60	74.7 ± 0.2 _(5.0_ _μg/mL)_	82.4 ± 3.2
Quercetin	0.47 ± 0.03	0.90 ± 0.04	-	99.73 ± 0.03	75.6 ± 0.1 _(1.5_ _μg/mL)_	-
Rutin	1.68 ± 0.27	-	-	80.41 ± 0.05	-	-
Gallic acid		1.02 ± 0.14	-	-	93.0 ± 0.1 _(1.75_ _μg/mL)_	-
Catechin		1.62 ± 0.10	-	-	87.5 ± 0.1 _(2.75_ _μg/mL)_	-
Ascorbic acid			1.95 ± 0.07			82.2 ± 3.2

IC_50_, concentration value at which 50% of the activity evaluated is reached; Max. Act., maximum activity expressed in percentage. The values are expressed as means ± (S.E.M) of three different experiments (*n* = 3). Max. Act. in the ABTS assay is considered for 5.0 μg/mL and at 250 μg/mL for the DPPH assay. “-“ indicates that the substance was not evaluated for the assay.

**Table 2 foods-11-02236-t002:** Relative Antioxidant Capacity (RAC).

	TFSR_AA_	TFSR_Rutin_	TFSR_Catechin_
IC_50_ (mg/mL)	0.0111 ± 0.0014	0.01201 ± 0.0013	0.01296 ± 0.0014
Maximum activity (%)	80.62 ± 2.63	76.40 ± 2.42	71.62 ± 2.81

TFSR_AA_, TFSR_Rutin_, and TFSR_Catechin_ represent the RAC of the TFSR relative to the maximum activity of ascorbic acid, rutin, and catechin, respectively. The values are expressed as means ± (S.E.M) of three different experiments (*n* = 3).

## Data Availability

The data used to support the findings of this study can be made available by the corresponding author upon request.

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
