# Peer review of "Advances in Evaluation of Antioxidant and Toxicological Properties of *Stryphnodendron rotundifolium* Mart. in *Drosophila melanogaster* Model"

_foods, 2022, doi:10.3390/foods11152236_

Round 1
Reviewer 1 Report
The paper „Advance in evaluation of Antioxidant and Toxicological Properties of Stryphnodendron rotundifolium Mart. in Drosophilamelanogaster model“ is a paper with some interesting results. It presents the properties of an extract and its toxicity. The results are interesting enough to publish. But the paper needs modifications before publishing. It is not clearly written. The results were explained on the basis of the individual active ingredients in the extract, but authors did not determine individual compounds in the extract. They just determined total flavonoids. The results need to be described according to the specific data in the paper. I would suggest explaining the results on the basis of the data from the paper, and after that compare to the data from literature and mention which individual compounds are present in the extract according to other studies. Some examples of the modifications are suggested:
L160 – please check the Abs sample blank. Is it maybe Abs fraction blank? Also check the equation at L159
L181 – please check the concentrations?
L239-249 – the sentence is not clear. Please rewrite. Please be careful when writing the names of compounds
Table 1 – please check the Table 1 and correct. Units need to be visible in the table, english needs to be corrected, the table needs to be clear when reading the title and other information for the first time.
L274-276 - „As shown in a previous study, the IC50 values of 5.4 and 12.0 μg/mL for the hydroalcoholic and aqueous extracts of S. rotundifolium Mart bark were in a similar trend to that reported by [8].“ The sentence should be clear, which previous study? Is it reference 8? Please write clear sentences
L285 – „…71 to 80%...“ Is it 72 to 81 %? Please be precise when describing your results
L289 – 294 „Evidences have shown that flavan-3-ol with catechol or pyrogallol groups at the B ring, in the presence of π-conjugated structures allow electronic shift and provide greater capacity to reduce metal ions and scavenge DPPH● radicals that can be stabilized by resonance to generate semiquinones and quinones, through electron donation and proton release [26-29]. Thus, the extracts inhibited DPPH● and ABTS•+ radicals as well as reduced Mo6+ and Fe3+ ions, due to the release of protons and electrons by polymeric PAs and flavan-3-ol present in TFSR“. This is based on the characterization of extracts by some other authors. Authors in this work only determined total flavonoids. They do not know which individual compounds are present in the extracts and therefore it is better not to base the conclusions according to literature. Some corrections of the text are suggested.
Figure 1 „Letters a, b, c, d, e, f, represent concentrations of substances“. Are you sure those are concentrations? If they are concentrations, which concentrations? Please correct if needed.
L322-323 „TFSR exhibited inhibition of 60.58 ± 4.74%, statistically similar to the catechin control (53.62 ± 6.17%),…“ Statistical similarity cannot be seen from the Figure. Please check the rest of the text describing Figure 1 and correct if necessary
L344-346 – „Therefore, TFSR could be directed to the development of bio-products that help to combat oxidative damage mediated by iron overload.“ Please be careful when making claims. TFSR might be suggested for the development of…Apply this for the rest of the text
L346-350 – „The significant free radical scavenging activity, metal ion reduction and inhibition of lipid peroxidation by TFSR is related to synergistic effects of prodelphinidins, pro-robinetidins and procyanidins with catechol and galloyl groups, indicating that S. rotundifolium Mart. possess the potential to be used in the development of target drugs for the treatment of diseases associated with Fe overload and oxidative stress mediated diseases“. Rewrite this sentence. The individual compounds were not determined in this paper. So these claims need to be written in a different way.
L422-424 – „Furthermore, these compounds can be easily absorbed from the gastrointestinal tract, which is a desirable feature for substances used in the development of oral drugs [44].“ Please check if this is true
Author Response
RESPONSE TO REVIEWERS 1
Dear Editor/ Reviewer,
Submission of revised manuscript "Advances in evaluation of Antioxidant and Toxicological Properties of Stryphnodendron rotundifolium Mart. in Drosophila melanogaster model."
On behalf of the authors, I sincerely appreciate the Editor and Reviewers for their criticism, observation and contributions towards improving the quality of our manuscript Advances in evaluation of Antioxidant and Toxicological Properties of Stryphnodendron rotundifolium Mart. in Drosophila melanogaster model."
I am herewith submitting our revised version of the manuscript having answered the queries raised and comments as suggested by the reviewers. The texts and bibliography as well as the tables and figures have been edited in line with the reviewer’s suggestion and queries have been well addressed to the best of our ability. The changes made on the texts are now tracked and marked up in the revised version of the manuscript. Thank you
Sincerely Yours,
Gerson Javier Torres Salazar; ORCID: 0000 – 0003 – 0908 – 0919; timotygertor@yahoo.com; Department of Biological Chemistry, Natural Products Research Laboratory, Regional University of Cariri, Av. Cel. Antônio Luiz, 1161. Pimenta, 63105-000 Crato-CE, Brasil. Tel.: (+55) 88-996985037.
Author's Reply to the Review Report (Reviewer 1)
Comments and Suggestions for Authors
The paper “Advance in evaluation of Antioxidant and Toxicological Properties of Stryphnodendron rotundifolium Mart. in Drosophila melanogaster model” is a paper with some interesting results. It presents the properties of an extract and its toxicity. The results are interesting enough to publish. But the paper needs modifications before publishing. It is not clearly written. The results were explained on the basis of the individual active ingredients in the extract, but authors did not determine individual compounds in the extract. They just determined total flavonoids. The results need to be described according to the specific data in the paper. I would suggest explaining the results on the basis of the data from the paper, and after that compare to the data from literature and mention which individual compounds are present in the extract according to other studies. Some examples of the modifications are suggested:
The present work is part of the sequence of a developed previously study with the TFSR (cited in the present work as reference number 9), in which the presence of individual phenolic compounds and quantification of total polyphenols, hydrolyzable and condensed tannins were determined.
L160 – please check the Abs sample blank. Is it maybe Abs fraction blank? Also check the equation at L159
- Answered in the manuscript
L181 – please check the concentrations?
- Answered in the manuscript
L239-249 – the sentence is not clear. Please rewrite. Please be careful when writing the names of compounds
- Answered in the manuscript
Table 1 – please check the Table 1 and correct. Units need to be visible in the table, english needs to be corrected, the table needs to be clear when reading the title and other information for the first time.
- Answered in the manuscript
L274-276 - “As shown in a previous study, the IC50 values of 5.4 and 12.0 μg/mL for the hydroalcoholic and aqueous extracts of S. rotundifolium Mart bark were in a similar trend to that reported by [8].” The sentence should be clear, which previous study? Is it reference 8? Please write clear sentences
- Answered in the manuscript
L285 – „…71 to 80%...“ Is it 72 to 81 %? Please be precise when describing your results
- Answered in the manuscript
L289 – 294 “Evidences have shown that flavan-3-ol with catechol or pyrogallol groups at the B ring, in the presence of π-conjugated structures allow electronic shift and provide greater capacity to reduce metal ions and scavenge DPPH● radicals that can be stabilized by resonance to generate semiquinones and quinones, through electron donation and proton release [26-29]. Thus, the extracts inhibited DPPH● and ABTS•+ radicals as well as reduced Mo6+ and Fe3+ ions, due to the release of protons and electrons by polymeric PAs and flavan-3-ol present in TFSR”. This is based on the characterization of extracts by some other authors. Authors in this work only determined total flavonoids. They do not know which individual compounds are present in the extracts and therefore it is better not to base the conclusions according to literature. Some corrections of the text are suggested.
- Answered in the manuscript
Figure 1 “Letters a, b, c, d, e, f, represent concentrations of substances”. Are you sure those are concentrations? If they are concentrations, which concentrations? Please correct if needed.
- Answered in the manuscript
L322-323 “TFSR exhibited inhibition of 60.58 ± 4.74%, statistically similar to the catechin control (53.62 ± 6.17%),…” Statistical similarity cannot be seen from the Figure. Please check the rest of the text describing Figure 1 and correct if necessary
- Answered here
The superposition of the error bars of the catechin control and the TFSR (0.050 mg/mL) may demonstrate the statistical similarity
L344-346 – “Therefore, TFSR could be directed to the development of bio-products that help to combat oxidative damage mediated by iron overload.” Please be careful when making claims. TFSR might be suggested for the development of…Apply this for the rest of the text
- Answered in the manuscript
L346-350 – “The significant free radical scavenging activity, metal ion reduction and inhibition of lipid peroxidation by TFSR is related to synergistic effects of prodelphinidins, prorobinetidins and procyanidins with catechol and galloyl groups, indicating that S. rotundifolium Mart. possess the potential to be used in the development of target drugs for the treatment of diseases associated with Fe overload and oxidative stress mediated diseases”. Rewrite this sentence. The individual compounds were not determined in this paper. So these claims need to be written in a different way.
- Answered in the manuscript
L422-424 – “Furthermore, these compounds can be easily absorbed from the gastrointestinal tract, which is a desirable feature for substances used in the development of oral drugs [44].“ Please check if this is true
Answered in the manuscript
Reviewer 2 Report
Dear authors,
I have revied your paper Advance in evaluation of Antioxidant and Toxicological Properties of Stryphnodendron rotundifolium Mart. in Drosophila melanogaster model.
1. Typos and minor errors:
· L32 in vivo
· L611 – DOI is missing
· p italic L513, 518 … and everywhere through text
· formulas font is not correct
2. You should add a scheme to categorize your work to be easier to follow
3. English should be improved
4. In the discussion part, I am missing the bigger picture of your paper. Add a few sentences about possible real application
Author Response
RESPONSE TO REVIEWERS 2
Dear Editor/ Reviewer,
Submission of revised manuscript "Advances in evaluation of Antioxidant and Toxicological Properties of Stryphnodendron rotundifolium Mart. in Drosophila melanogaster model."
On behalf of the authors, I sincerely appreciate the Editor and Reviewers for their criticism, observation and contributions towards improving the quality of our manuscript "Advances in evaluation of Antioxidant and Toxicological Properties of Stryphnodendron rotundifolium Mart. in Drosophila melanogaster model."
I am herewith submitting our revised version of the manuscript having answered the queries raised and comments as suggested by the reviewers. The texts and bibliography as well as the tables and figures have been edited in line with the reviewer’s suggestion and queries have been well addressed to the best of our ability. The changes made on the texts are now tracked and marked up in the revised version of the manuscript. Thank you
Sincerely Yours,
Gerson Javier Torres Salazar; ORCID: 0000 – 0003 – 0908 – 0919; timotygertor@yahoo.com; Department of Biological Chemistry, Natural Products Research Laboratory, Regional University of Cariri, Av. Cel. Antônio Luiz, 1161. Pimenta, 63105-000 Crato-CE, Brasil. Tel.: (+55) 88-996985037.
Author's Reply to the Review Report (Reviewer 2)
Comments and Suggestions for Authors
I have revied your paper Advance in evaluation of Antioxidant and Toxicological Properties of Stryphnodendron rotundifolium Mart. in Drosophila melanogaster model.
- Typos and minor errors:
- L32 in vivo
- Answered in the manuscript
- L611 – DOI is missing
- Answered in the manuscript
- p italic L513, 518 … and everywhere through text
- Answered in the manuscript
- Formulas font is not correct
- Answered in the manuscript
- You should add a scheme to categorize your work to be easier to follow
- Answered in the manuscript
- English should be improved
- Answered in the manuscript
- In the discussion part, I am missing the bigger picture of your paper. Add a few sentences about possible real application
- Answered in the manuscript
